# Inflammation and Perioperative Cardiovascular Events

**DOI:** 10.3390/cells14171362

**Published:** 2025-09-01

**Authors:** Peter Poredos, Radko Komadina

**Affiliations:** 1Department of Anaesthesiology and Surgical Intensive Care, University Medical Centre Ljubljana, Zaloska 7, 1000 Ljubljana, Slovenia; peter.poredos@kclj.si; 2The Faculty of Medicine, University of Ljubljana, Vrazov Trg 2, 1000 Ljubljana, Slovenia; 3Surgical Facilities, Campus Celje, The Faculty of Medicine, University of Ljubljana, Oblakova 5, 3000 Celje, Slovenia

**Keywords:** perioperative inflammation, cardiovascular events, inflammatory markers, prevention of perioperative systemic inflammation

## Abstract

Surgery and anesthesia induce a stress response that provokes increased sympathetic stimulation, secretion of cortisol, hypercoagulability, and systemic inflammatory response. All these homeostatic deteriorations, especially systemic inflammation, represent a risk for organ damage. Perioperative cardiac complications have an increasing impact on morbidity and mortality, not only in cardiovascular but also in non-cardiac surgery. Surgical procedures represent a potential trigger for systemic inflammation that causes secretion of proinflammatory cytokines, activation of neutrophils, and tissue damage. Also, increased levels of preoperative inflammatory markers predict perioperative cardiovascular events. Systemic inflammatory biomarkers increase during the first days after surgical procedures and decline within a few weeks. Besides contemporary traditional biomarkers (CRP, BNP), newer biomarkers, such as galectin-3, TNF-α, and various MiRNAs, can predict inflammatory response and related cardiac injury. Determination of inflammatory markers in the perioperative period could help identify patients at risk for cardiovascular events. The reduction in perioperative inflammatory response may improve surgical outcomes. Prevention and treatment of systemic inflammation can be achieved by optimization of surgical procedures, anesthetic regimen, and pharmacological agents, especially interleukin inhibitors. Determination of inflammatory biomarkers, along with prevention and treatment of inflammation, can improve perioperative cardiac risk reduction strategies.

## 1. Introduction

The immune–inflammatory system represents a protective mechanism that defends the organism against harmful environmental influences, especially infections. Besides infection, an inflammatory response may also be triggered by tissue damage, including surgical procedures [1]. Tissue damage that occurs during surgery provokes inflammation, which is a primary promoter of tissue’s natural repair and healing. However, an adequate balance between pro- and anti-inflammatory processes within the innate and adaptive immune system is essential [2]. On the other hand, this regulated hyperinflammatory response may increase the risk of postoperative complications that contribute to poorer quality of recovery and lead to a persistent disability or death [3]. Cardiovascular (CV) complications after major surgery represent one of the main risks of perioperative morbidity and mortality [4]. Surgical procedures trigger perioperative CV complications through the provocation of systemic inflammation [5].

This narrative review aims to elucidate the involvement of inflammation in perioperative CV events and to speculate whether the determination of inflammatory biomarkers could improve perioperative cardiac risk reduction strategies. Further, the effect of perioperative anti-inflammatory strategies for the prevention of CV events is discussed.

## 2. Pathophysiology of Surgical Inflammation

The inflammatory response to invasive procedures is an inherent answer of the body to the intervention and can be both beneficial and potentially harmful. Tissue damage that occurs during surgery triggers defense mechanisms, including immuno-inflammatory reactions [5]. Tissue injury leads to the release of damage-associated molecular patterns (DAMPs), also called alarmins. Alarmins bind to receptors of a variety of endothelial and inflammatory cells and activate them. Activation of these cells triggers the expression of adhesion molecules that facilitate leukocyte migration and the release of inflammatory mediators on the local level [6]. However, an excessive perioperative inflammatory response provokes systemic inflammation involving endothelial cell activation, glycocalyx damage, additional inflammatory mediator release, and immune cell activation [7]. Aside from inflammatory response caused by tissue trauma, surgery also carries a high risk for perioperative infections, which lead to inflammatory complications [8]. Clinical presentations of infectious and noninfectious causes of systemic perioperative inflammation overlap considerably; however, the differentiation between surgery-related inflammation and infection represents an important step in the postoperative management of inflammation. Inflammation after surgical injury is characterized by accumulation of inflammatory cells, damage of endothelial cells, and upregulation of inflammatory mediators.

### 2.1. Activation and Accumulation of Inflammatory Cells

After surgical injury, polymorphonuclear leukocytes, macrophages, and lymphocytes become activated by the secretion of various mediators. Leucocytes interact with activated endothelial cells through E-selectins expressed on endothelial cells. Neutrophils further amplify inflammatory response through myeloperoxidase (MPO), facilitating the recruitment of additional neutrophils [9]. Macrophages release chemokines and stimulate phagocytosis of apoptotic neutrophils and contribute to perioperative inflammation [10].

Platelets primarily provide hemostasis by limiting bleeding in case of vascular damage. However, they are also involved in the perioperative inflammatory process [11]. Platelets modulate the secretory potential of neutrophils and regulate barrier integrity [12]. As surgery provokes the activation of platelets, it thus affects not only the hemostatic performance, but also the inflammatory response.

### 2.2. Damage to Endothelial Cells

Inflammatory cell recruitment, cytokines, and the release of ROS influence endothelial cell function and integrity. Proinflammatory cytokines promote endothelial cell activation, which triggers neutrophil and platelet recruitment and initiates the expression of adhesion molecules [13]. Endothelium is not only a barrier that protects from inflammatory cell infiltration, but also a sensor for systemic inflammation in the perioperative period.

Surgical procedures can also damage glycocalyx, adding sugar-like proteoglycan coating to endothelial cells. The glycocalyx protects the endothelium from interaction with adhesion of leucocytes and platelets [14]. Glycocalyx disruption causes endothelial cells to switch to a proinflammatory phenotype characterized by increased intracellular adhesion molecule-1 (ICAM-1).

### 2.3. Release of Inflammatory Mediators

Surgical damage to the skin plays an important role in the activation of inflammation. Following the skin and subcutaneous fat incision, a neutrophil influx occurs, which is accompanied by an increase in release of inflammatory markers [15]. Incised subcutaneous fat promotes inflammation and postsurgical infections because adipocytes release tumor necrosis factor alpha (TNF-α), which triggers an inflammatory response [16]. In the mouse model, it was shown that bleeding itself may also trigger an immune reaction [17]. Further, reperfusion injury, which occurs after clamping of blood vessels, leads to the production of reactive oxygen species (ROS) and neutrophil influx into the ischemic organ [18]. Hypoxemia also amplifies the inflammatory state by upregulation of chemokines and cytokines. Even minimally invasive surgical procedures cannot altogether abolish mediator release [19].

After breaking the skin, chemokines are released, followed by a release of macrophage inflammatory protein-2 (MIP-2) [20]. Cytokines are key modulators of inflammation and play both inflammatory and anti-inflammatory roles [21]. The balance between pro- and anti-inflammatory cytokines affects immunity and infection as well as wound healing. Cytokine release is regulated by mitogen-activated protein kinase/extracellular signal- regulated kinase (MAPK/ERK). Protein kinases are produced by the cells themselves through the process of gene expression [22]. MAPKs, through regulation of cytokine production, stimulate endothelial activation vascular dysfunction and myocardial damage.

During and after major surgery (especially in patients with cardiovascular comorbidities), an activated systemic inflammatory response can contribute to endothelial damage and coagulation imbalance. The immune system acts as a sentinel in the human body to provide protection from infection and regulate inflammatory responses. Phosphatidylinositol-4,5 bisphosphate 3-kinase catalytic subunit-β (PI3K-β) and mechanistic target of rapamycin (mTOR) play a crucial role in various cellular functions, immunity, and inflammation. The mTOR pathway is dysregulated in several diseases, particularly in some cancers, infections, and tissue damage [23]. Activation of the PI3K-AKT signaling pathway can inhibit the inflammatory response and apoptosis after tissue injury, including surgical procedures [24].

## 3. Time Course of Circulating Inflammatory Markers After Surgical Procedure

Several circulating inflammatory mediators, pro- and anti-inflammatory cytokines, including tumor necrosis factors, T-cells, transforming growth factor (TGF)-β, and macrophage migration inhibitory factor (MIF), have been considered as central mediators of inflammatory response immediately after a surgical procedure [25]. The levels of several cytokines generally peak immediately after the surgery and correlate with the duration and the extent of surgery. The increase in cytokine levels is mainly independent of preoperative levels [26]. Cytokines provoke inflammatory cell recruitment and modulate the activation of neutrophils and macrophages. Also, additional biomarkers, like C-reactive protein (CRP), increase in the first week after major surgery [27]. The peak of CRP values after orthopedic surgery was observed in 2–3 days after surgery, and the secondary increase in CRP levels may indicate the risk of complications [28]. In one of the studies, including patients undergoing total hip replacement in the postoperative period, inflammatory markers increased. Interleukin-6 (IL-6) significantly increased a day after surgery, and after 5 days, a significant decrease was seen. After surgery, leucocyte count, neutrophil/lymphocyte ratio (NLR), and platelet/lymphocyte ratio (PLR) increased and remained significantly higher than before surgery 5 days after the procedure [29]. The neutrophil/lymphocyte ratio is an inflammatory marker, predicting postoperative infectious complications [30]. Recently, newer biomarkers, like galectin-3, sST-2, GDF-15, MiRNAs, and many other markers which increase immediately after a surgical procedure and predict inflammation were identified [31].

## 4. Perioperative Systemic Inflammation and Cardiovascular Events

Preoperative systemic inflammation, as well as surgically provoked inflammatory response, is related to CV events. Perioperative CV complications have an increasing impact on morbidity and mortality of patients requiring major surgery. Myocardial injury after non-cardiac surgery has an estimated incidence of 20% and has an equally significant impact on morbidity and mortality as myocardial infarction outside of the perioperative period. Myocardial injury after surgery is associated with an increased risk of short- and long-term mortality compared to patients undergoing surgery without myocardial injury (one-year mortality rate 22.5% vs. 9.3%) [32]. Approximately 20% of patients having non-cardiac surgery develop CV complications within 30 days [33]. Besides hemodynamic deterioration and oxygen supply mismatch, perioperative inflammation influences postoperative adverse cardiac events [34].

### 4.1. Preoperative Inflammatory Disease and CV Events

It is well known that patients preoperatively diagnosed with inflammatory disease, such as rheumatoid arthritis or inflammatory bowel disease, are at an increased risk of developing CV complications [35]. Specifically, preoperative hsCRP is a strong predictor of postoperative CV complications. The study of Ackland et al. showed that preoperatively elevated systemic inflammation, as reflected by elevated preoperative neutrophil-lymphocyte ratio (NLR), predisposes patients to perioperative myocardial injury. Myocardial injury was more common in patients with preoperative NLR > 4 [36]. In patients with carcinoma, NLR is a key determinant of outcome. Increased NLR is a marker of more advanced and aggressive disease (increased stage, number of metastatic lesions) and a predictor of poor outcome in a variety of cancers with increased 30-day mortality, increased postoperative infections, and stay in the intensive care unit [37].

Furthermore, preoperative activated inflammatory status, defined by CRP and fibrinogen in patients who undergo cardiac surgery, affects long-term outcomes and is an independent risk factor of mortality [38]. Thus, preoperative inflammatory disease can increase the risk of perioperative CV complications, which can manifest as myocardial infarction, arrhythmia, heart failure, and other adverse CV events.

### 4.2. Surgically Provoked Inflammation and CV Events

In patients without preoperative inflammatory disease, surgery and anesthesia increase the risk of perioperative CV complications, as the stress response caused by sympathetic stimulation provokes a release of inflammatory cytokines. Inflammation can lead to volume shifts and reduce the kidney function in the postoperative period and thus affect cardiac preload [39]. Postoperative myocardial strain due to inflammation leads to subclinical or clinical heart failure, shown by proBNP increase [40]. Myocardial injury after non-cardiac surgery occurs in up to 18% of patients over 45 years of age [41], and 4% of patients with myocardial injury after surgery will die within 30 days after the procedure [42]. After cardiac surgery, the incidence of postoperative systemic inflammation was assessed in the study of Squiccimarro et al., and the association between postoperative systemic inflammation and postoperative outcomes was investigated. Out of 502 patients who underwent cardiac surgery, 142 patients (28.3%) developed systemic inflammatory response syndrome (SIRS), and SIRS was associated with more complicated postoperative course and higher postoperative morbidity [43].

### 4.3. Relationship Between Intensity of Perioperative Inflammatory Response and CV Events

In one of the studies, in patients who underwent cardiac surgery, SIRS was registered in 48.3% of patients on the first postoperative day. On the fourth day after surgery, it declined to 15.7%. In patients with prolonged SIRS, atrial fibrillation appeared in 42.2% of patients, and SIRS was an independent risk factor for postoperative atrial fibrillation [44]. A more severe form of SIRS during the first four postoperative days has been reported to correlate with high risk of multiple organ dysfunction, CV events, and postoperative mortality [45]. A large meta-analysis on 29,401 patients undergoing cardiac surgery showed a significant correlation between preoperative CRP levels and major adverse CV events [46].

In the review of 52 studies with a total of 121,849 patients who underwent non-cardiac surgery, inflammatory biomarkers were measured preoperatively and within 10 days after surgery. Outcomes included all-cause mortality and major adverse CV events. Inflammatory biomarker levels (CRP, fibrinogen, and IL-6) in the perioperative period were associated with all-cause mortality and adverse CV events in patients undergoing non-cardiac surgery [47]. Patients with increased preoperative plasma levels of IL-6 had higher preoperative concentrations of cardiac biomarkers, suggesting the presence of subclinical cardiovascular disease [48]. Despite this, the incidence of perioperative CV events was not significantly higher in this patient group.

Changes in biomarker levels from the preoperative to the postoperative period could be a better predictive measure of postoperative outcomes. In an analysis of the Assessment of Clinical Effects of Cholesteryl Ester Transfer Protein Inhibition With Evacetrapib (ACCELERATE) study, which enrolled 12,092 patients with high-risk vascular disease, longitudinal changes in hsCRP improved the prediction of major CV events compared to the baseline highest CRP. All these data support the hypothesis that perioperative myocardial injury and other CV complications are related to activated systemic inflammation pre- or postoperatively [49]. Recently, a Modified Glasgow prognostic score (mGPS), a scoring system for the prediction of periprocedural adverse events, including CV events, has been studied as a risk assessment tool for perioperative CV complications in patients with preoperative inflammation. It incorporates preoperative CRP values. However, its value in the prediction of perioperative CV events in different surgeries has not been sufficiently studied [50] (Table 1).

## 5. Risk Factors for Postoperative Inflammatory Response and the Influence of the Type of Surgical Procedure

Systemic inflammatory response to cardiac and major non-cardiac surgery is known to vary substantially between individual patients, both at clinical and biochemical levels [53]. Patients who develop a more severe systemic inflammatory response might have an increased risk of adverse outcomes [54]. The causes for variation in inflammatory response are multifactorial and are probably a result of genetics and the intensity of perioperative stimuli, which activate a systemic immune response. In the study of Dieleman et al., increased patient age was associated with a lower prevalence of postsurgical SIRS. This finding supports the hypothesis that with aging, patients have reduced immune response [55]. Further, obesity, diabetes, and preexisting CV conditions increase the risk of an inflammatory response in perioperative period. Additionally, certain medications and supportive treatment, such as blood transfusion, can exacerbate the inflammatory response [56]. Perioperative inflammation also depends on utilization of extracorporeal support (ECMO) and perioperative mechanical ventilation, especially on high tidal volume ventilation [57].

### 5.1. Type of Surgical Procedure and Inflammatory Response

Surgical factors, such as prolonged surgical time, extensive surgical procedure and the presence of infection can also contribute to inflammatory response. Further, perioperative inflammatory response depends on the type of surgical procedure. The range of cytokine release and their concentration reflect the magnitude of surgical trauma. Laparoscopic surgery decreases local and systemic production of cytokines and acute-phase reactants and better preserves immunity compared to open abdominal surgery [58]. An experimental study demonstrated a reduced hypersensitivity response (reduced release of oxygen-free radicals and TNF-α) for the laparoscopic approach compared to conventional open surgery [59]. Further, the comparison between conventional laparoscopic transabdominal preperitoneal hernia repair and robot-assisted surgery showed lower CRP and IL-6 levels and fewer perioperative complications in the robot-assisted procedure [60]. In contrast to these findings, a multicenter randomized controlled trial comparing perioperative inflammatory response between laparoscopic and open pancreatodoudenectomy showed that the laparoscopic procedure did not reduce the postoperative inflammatory response [61]. The explanation for these contradictory results could be the consequence of the type of surgical procedure. For both surgical techniques, pancreatoduodenectomy is a highly complex and a long-lasting procedure and the benefit of laparoscopy could be insufficient to compensate for the extent of the procedure. A systematic review and a meta-analysis compared the humoral inflammatory response after laparoscopic and open colorectal cancer resection [62]. Circulating levels of CRP, IL-6, IL-8, TNF-alpha, and endothelial growth factor (VEGF) were followed. A total of 1131 patients with colorectal cancer resection were included. The increase in postoperative levels of several proinflammatory markers, particularly IL-6, IL-8, and TNF-alpha, was significantly less pronounced after laparoscopic than open surgical procedure. Further, surgical stress and inflammatory response were compared between patients with a robot-assisted and a laparoscopic surgery of colon cancer [63]. Robot-assisted surgery was associated with a reduction in CRP on the first postoperative day. However, no statistically significant differences were noted for CRP expression on the second and third postoperative days.

### 5.2. Perioperative Inflammation and Cancer Cell Seeding

Perioperative inflammatory response may also contribute to cancer recurrence and metastasis. Immunosuppression caused by systemic inflammation inhibits natural killer cells and T-lymphocyte activity, which is critical for the clearance of circulating tumor cells [64]. Clinical trials have shown that minimally invasive video-assisted thoracoscopic surgery attenuates the inflammatory response (lower IL-6 and CRP) and maintains immune cell functions compared to open thoracotomy [65]. This approach can prevent seeding of cancer cells. Video-assisted thoracoscopic surgery also resulted in significantly lower levels of IL-6 in pleural fluid following lobectomy compared to open lobectomy [66]. Therefore, acute inflammation triggered by surgery for malignant tumors can suppress immune cells that kill tumor cells [64].

## 6. Prevention of Perioperative Inflammatory Response and Cardiovascular Events

Perioperative CV complications are a source of morbidity and mortality for more than 200 million patients worldwide who undergo non-cardiac and cardiac surgery each year. Therefore, the question arises, how to reduce or prevent the inflammatory response [67]. Recent investigations indicate that protective strategies to prevent excessive inflammatory response may improve patients’ outcomes [5].

Antinflammatory drugs: Besides optimization of surgical procedures, pharmacologic agents, such as tocilizumab, IL-6 receptor antibody, and steroids, attenuate the inflammation and have been shown to improve postoperative clinical outcome. Tocilizumab has been shown to modulate inflammation in a rat model and in patients with severe coronavirus disease [68]. Meta-analysis of studies assessing the effects of corticosteroids in patients with septic shock did not show a significant reduction in mortality [69]. Substances protecting the endothelial glycocalyx have been shown to modulate systemic inflammation. Heparanase-2 protects against lipopolysaccharides-mediated endothelial damage and inflammatory response [70]. However, the data on the efficacy and clinical applicability of these new inflammatory inhibitors are limited.

Aspirin is a potent inhibitor of cyclooxygenase-1, preventing platelet aggregation and mitigating thrombotic risk at the cost of increased bleeding. Further, it also has an anti-inflammatory effect. However, in the Perioperative Ischemic Evaluation (POISE-2) trial, which randomly assigned 10,010 patients at risk for CV complications after non-cardiac surgery, no benefit of aspirin was observed at 30 days in any subgroup analysis. Its use was associated with a higher incidence of major bleeding [71].

Colchicine presents anti-inflammatory properties through the reduction in IL-6 and hsCRP and was shown to significantly reduce the incidence of major CV events [72]. Colchicine can inhibit various leukocyte functions, which represents its most significant anti-inflammatory effect. Colchicine was shown to reduce the incidence of postoperative atrial fibrillation and post-pericardiotomy syndrome [72]. Canakinumab Anti-inflammatory Thrombosis Outcome Study (CANTOS) [73] showed that monoclonal antibodies targeting IL-1β (a cytokine that drives the IL-6 signaling pathway) significantly reduced recurrent major CV events. Further, direct targeting of IL-1β has become a priority for prevention of perioperative CV events. Recently, the efficacy of different IL-1β antagonists like canakinumab and ziltivekimab in the prevention of perioperative CV events has been investigated [74].

Lipid lowering therapy with statins: It is a promising approach to reduce perioperative CV events, which is most probably based on their pleiotropic–anti-inflammatory effects. It is well known that statins can reduce the circulating levels of inflammatory markers and slow the inflammatory process [75]. A retrospective analysis of 204,885 patients undergoing non-cardiac surgery demonstrated that patients taking lipid-lowering agents in the first 2 days of hospitalization had a significantly reduced in-hospital mortality [76]. The meta-analysis of controlled trials in vascular surgery—Dutch Echocardiographic Cardiac Risk Evaluation Applying Stress Echo III (DECREASE-III) study showed that perioperative treatment with fluvastatin was associated with a 53% reduction in death or myocardial infarction [77]. However, the Cochrane review of randomized controlled trials of statins in unselected non-cardiac surgery reported that there is insufficient evidence that perioperative statins reduce CV events [78]. ACC/AHA guidelines on perioperative CV evaluation and management of patients undergoing non-cardiac surgery recommend continuing clinically indicated statin use in the perioperative period of non-cardiac surgery. Initiation of statin therapy before surgery should be considered for patients undergoing CV surgery (Class IIa recommendation) [79].

Other strategies: Anesthetic strategies may play a role in perioperative inflammatory response. A volatile anesthetic, sevoflurane, reduces neutrophil apoptosis in a mouse model [80], and propofol inhibits human neutrophil function [81]. Because high tidal volume ventilation may also trigger an inflammatory response, adjusting ventilation strategies in anesthetized patients may also affect perioperative inflammation. Other surgical strategies, including minimizing tissue trauma, shortening surgical duration, and utilization of support system (ECMO), are also modulators of perioperative inflammatory response [5] (Table 2).

## 7. Discussion and Conclusions

Systemic inflammation represents a risk for perioperative CV and other adverse events. Therefore, patients with preoperative inflammatory disease are at increased risk for perioperative complications. However, also surgical procedures represent a potential trigger of systemic inflammation based on secretion of proinflammatory cytokines, activation of neutrophils and endothelial dysfunction. Perioperative systemic inflammatory response depends on different factors, including duration and the type of surgery, intensity of tissue trauma, blood loss, anesthetic regimen, and underlying diseases. In non-cardiac surgery, myocardial damage, infarction, and atrial fibrillation represent the most frequent CV events. For patients at high risk for postoperative CV complications, it is important to reduce inflammatory response and related risks and optimize underlying cardiac conditions. Preventive and therapeutic options include optimization of surgical procedures, anesthetic regimens, and pharmacologic agents. Recent studies indicated that minimally invasive laparoscopic and robot-assisted surgical procedures are associated with lower inflammatory response and perioperative events than open surgery. However, their applicability is limited.

As inflammation represents one of the basic pathogenetic mechanisms of perioperative complications, it is helpful to identify patients in whom excessive perioperative inflammatory response is expected. Inflammatory markers (CRP, IL-6, and white blood cell count) are increasingly recognized as valuable tools in perioperative risk assessment, providing an insight into a patient’s baseline inflammatory status and their potential to develop postoperative complications. Despite this fact, routinely used scoring systems for perioperative risk assessment (ACS NSQIP, P-POSSUM, CR-POSSUM, EuroSCORE II, ASA, RCRI) do not include preoperative levels of any inflammatory markers. Few recent studies have shown that the predictive power of current clinical risk evaluation systems could be significantly strengthened by inflammatory biomarkers [82]. Consequently, novel and supplementary preoperative risk indices have been developed; however, they have not been validated yet. Incorporation of inflammatory markers into preoperative risk assessment tools might support delaying surgery for additional patient optimization, support perioperative statin use and anti-inflammatory agents (colchicine), or indicate more aggressive patient monitoring.

Early detection and adequate treatment of systemic inflammation could result in improved outcomes after surgery.

## Figures and Tables

**Table 1 cells-14-01362-t001:** A relationship between inflammatory markers and perioperative cardiovascular events.

Inflammatory Marker	Clinical Relevance
hsCRP	- Preoperative hsCRP is a strong predictor of postoperative CV complications (MI, arrhythmia, HF) [35,46] - Longitudinal changes in hsCRP improve the prediction of major CV events [49]
IL-6	- Increased IL-6 represents a higher risk of CV death, MACE, MI, stroke, peripheral artery disease, and HF [51] - Increased preoperative IL-6 is related to higher levels of cardiac biomarkers, suggesting the presence of subclinical CV disease [48]
NLR	Preoperative NLR > 4 presents a higher risk for perioperative myocardial injury [36]
mGPS	mGPS could be a useful risk assessment tool for perioperative CV events in patients with preoperative inflammation, but is not yet sufficiently studied [50]
Ferritin	Ferritin values ≥ 141 ng/mL might be used as a predictive postoperative atrial fibrillation biomarker in cardiac surgery [52]

CV—cardiovascular, HF—heart failure, hsCRP—high sensitivity C-reactive protein, IL-6—interleukin-6, MACE—major adverse cardiovascular events, mGPS—modified Glasgow prognostic score, MI—myocardial infarction, NLR—neutrophil/lymphocyte ratio.

**Table 2 cells-14-01362-t002:** Pharmacological agents modulating perioperative inflammatory response and related perioperative complications.

Drug	Study/Ref	Findings
IL-6 antibody - Tocilizumab	Tocilizumab in the treatment of severe coronavirus disease [68]	Tocilizumab provoked a significant decline in inflammatory markers and reduced ventilatory insufficiency
Aspirin	Perioperative Ischemic Evaluation (POISE-2) trial—10,010 pts. with noncardiac surgery [71]	In 30 days after surgery, no effects on perioperative CV events
Colchicine	Meta-analysis of 10 studies including patients undergoing cardiac surgery [72]	Colchicine significantly reduced postoperative atrial fibrillation and postpericardiotomy syndrome
Statins	Retrospective analysis of 204,885 pts. undergoing noncardiac surgery [76]	Significantly reduced in-hospital mortality
	Meta-analysis—DECREASE III study—perioperative treatment with fluvastatin of pts. undergoing major vascular surgery [77]	53% reduction in death or myocardial infarction
Anesthetic drugs - Propofol	In vitro study investigating effects of propofol on neutrophil function [81]	Propofol impaired neutrophil function
	Comparison of effects of propofol and sevoflurane on perioperative inflammatory response [80]	Propofol compared to sevoflurane significantly reduced perioperative inflammatory response in pts. undergoing lung cancer resection

CV—cardiovascular, IL-6—interleukin-6.

## Data Availability

No new data were created or analyzed in this study.

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
