# Peer review of "Inflammation and Perioperative Cardiovascular Events"

_cells, 2025, doi:10.3390/cells14171362_

Round 1

Reviewer 1 Report

Comments and Suggestions for Authors

The authors aimed to describe the role of inflammation in perioperative cardiovascular events and to explore whether assessing inflammatory biomarkers and their treatment could enhance strategies for reducing cardiac risk in the perioperative period. Overall, I found this review difficult to understand. Some concepts are unclear, and there are instances of duplicated descriptions. Additionally, the structure of the review is somewhat confusing.

 For example, Introduction: 39-40: “However, an inflammatory response may also be triggered by stimuli, such as surgical procedures. In this case an excessive response may lead to organ damage [1].” -in both cases, an excessive inflammatory response may lead to organ damage, not just surgical procedures. Line 42-44- which is essential and balances pro- and anti-inflammatory processes within the innate and adaptive immune system stimulating natural repair and healing: confused, please make it clear.

  1. Pathophysiology of surgical inflammation

It would be beneficial if the authors could provide subtitles to clearly state each stage of surgery caused inflammation.

  1. Surgical procedures and circulating inflammatory markers

There are no Surgical procedures and a mere list of inflammatory markers.

Other parts had similar issues.

Author Response

Dear Reviewer,

thank you for your meaningful and kind comments related to our manuscript.

Below please find point-by point response to the issues raised.

  1. According to your suggestions we tried to improve the content of our manuscript and made it more understandable. Therefore, Introduction was rewritten and sentences in the lines 39-44 were upgraded to more understandable versions,
  2. Chapter Pathophysiology was structured and additional information on the pathophysiology of inflammatory response was added.
  3. The title of paragraph 3 –»Surgical procedures and circulating inflammatory markers was substituted with a new one: “Time course of inflammatory markers after surgical procedure”.

Also, other parts of manuscript were corrected according to your suggestions.

We believe that your suggestions significantly improved the quality of our manuscript.

Reviewer 2 Report

Comments and Suggestions for Authors

Title: Inflammation and perioperative cardio-vascular events

In this paper, the authors study perioperative cardiac and cardiovascular complications, which can lead to stress, systemic inflammation, morbidity, and mortality. The authors conclude that the objective of this narrative review is to clarify the involvement of inflammation in perioperative cardiovascular events and to hypothesize whether determining and managing inflammatory biomarkers can improve strategies for reducing perioperative cardiac risk.

This paper adds something to the topic and is quite informative. However, I have some concerns.

Abstract:

TNF-alpha. Old fashion, remove alpha.

The authors conclude: "The goal of this narrative review is to clarify..." This is not a conclusion. The authors should comment on their findings, their experience.

The tables only contain the title and explanation of the acronyms. There is no legend that analyzes the data in the table.

For a better presentation:

I would add a figure.

This article lacks some molecular biology that would make it more interesting.

During surgery, the body experiences tissue trauma that induces a systemic inflammatory response. MAPKs regulate cytokine production, cell death, and endothelial activation and can contribute to myocardial damage, vascular dysfunction, and hemodynamic instability. In the light of these concepts, to make this paper more interesting for the readers of this important journal, the authors should expand a bit the discussion (or introduction). Below I report an interesting article that should be studied, incorporate the meaning and report it briefly in the discussion and in the list of references.

Saggini R, Pellegrino R. MAPK is implicated in sepsis, immunity, and inflammation. International Journal of Infection. 2024;8(3):100-104. (www.biolife-publisher.it).

During and after major surgery (especially in patients with cardiovascular comorbidities), a systemic inflammatory response is activated which can contribute to endothelial damage and coagulation imbalance. The β isoform (PI3Kβ) is expressed in platelets, endothelium, and immune cells, and activated by G-protein-coupled receptors (GPCRs), cytokines, and growth factors. Selective PI3Kβ inhibitors can prevent thrombosis without excessive bleeding risk. Again, here, we report  an article which has been recently published that should be studied, incorporate the meaning and report briefly in the discussion or introduction, and in the list of the references.

Avivar-Valderas A. Inhibition of PI3Kβ and mTOR influence the immune response and the defense mechanism against pathogens. International Journal of Infection. 2023;7(2):46-49. (www.biolife-publisher.it).

I believe these suggestions are important for improving this paper. Without these corrections the paper cannot be published. So I recommend minor revision.

Comments on the Quality of English Language

The quality of english language shoul be impoved 

Author Response

Dear Reviewer,

thank you for your meaningful and kind comments related to our manuscript.

Below please find point-by point response to the issues raised.

  1. The statement »The goal of this narrative review« was eliminated and substituted with more scientific meaning.
  2. According to your suggestions paragraph 2 –Pathophysiology of surgical inflammation was extended and information dealing with molecular biology including MAPKs and the role of beta-isoform PI3K was included.

We believe that your suggestions significantly improved the quality of our manuscript.

Reviewer 3 Report

Comments and Suggestions for Authors

This review covers in a concise manner how inflammation (both baseline and triggered by surgery or infection) may result in an elevated cardiovascular risk after surgical procedures.

It is generally well-written and represents a valuable overview of this field including a sufficiently high number of references. The included tables give a well-structured information on biomarkers used to assess the inflammatory state, as well as pharmacological compounds that are considered for modulating the inflammatory state in surgery patients.

There are only minor points that should be considered for a revised version of the manuscript:

  • Line 23: specify which miRNA would serve as biomarker or change wording to “various miRNAs”
  • Line 164: Change “One hundred and forty-two patients” to the number 142.
  • Line 178ff: Wording is not completely correct
  • Lines 308, 309 – check whether you really mean IL-6 and not IL-1

Further suggestions for improvements:

  • elaborate a bit more on the difference between acute and chronic inflammation (and put it in a context of local versus systemic inflammation).
  • Consider that acute inflammation, such as that triggered by surgery for malignant tumors, can activate immune cells that kill cancer cells. Therefore, acute inflammation may be beneficial, while chronic inflammation usually worsens cancer immune surveillance.
Comments on the Quality of English Language

There are minor typographic inconsistencies (additional spaces, some minor grammar issues, hyphens within a word etc.), as well as language style issues (which can be corrected by the editorial team).

Author Response

Dear Reviewer,

thank you for your meaningful and kind comments, which significantly improved the quality of our manuscript.. We have revised our manuscript according to your suggestions.

Below please find point-by point response to the issues raised.

  1. Written inconsistencies in line: 23, 164 and 178 were corrected according to your suggestions;
  2. This manuscript is dealing with systemic inflammatory response after surgical procedure. Therefore, local inflammation is only shortly mentioned.
  3. According to your suggestion information on the beneficial effects of acute inflammation on cancer immune surveillance was included in the existing text at the end of paragraph 5.

Thank you once again and we hope the manuscript will fulfil the criteria for publication.

Round 2

Reviewer 1 Report

Comments and Suggestions for Authors

The authors have enhanced the manuscript by adding additional clarifications and subheadings to the initial sections. However, the later sections still require similar refinements to ensure clarity, akin to the improvements made in the earlier parts (starting from 4. Perioperative systemic inflammation and cardiovascular events)

Author Response

Dear reviewer,

thank you very much for your thoughtful and constructive suggestions. We additionally enhanced also the rest of the chapters in the manuscript and added subheadings. Also, the rearrangement of the text has been performed as to make the text more redable and understandable.

We hope the manuscript fulfils the criteria for publication in your respected journal.

Round 3

Reviewer 1 Report

Comments and Suggestions for Authors

I have no further questions.